

# Early abolition of cough reflex predicts mortality in deeply sedated brain-injured patients

Stanislas Kandelman[1,2], Jérémy Allary[1], Raphael Porcher[3], Cássia Righy[4,5], Clarissa Francisca Valdez[4,6], Frank Rasulo[7,8], Nicholas Heming[9], Guy Moneger[9], Eric Azabou[10], Guillaume Savary[1], Djillali Annane[9], Fabrice Chretien[11], Nicola Latronico[7,8], Fernando Augusto Bozza[5,12], Benjamin Rohaut[13,14], Tarek Sharshar[11,12,15] and Groupe d'Exploration Neurologique en Réanimation (GENeR)

[1] Department of Anesthesiology and Intensive Care Unit, Beaujon Hospital, University Denis Diderot, Clichy, France
[2] Department of Anesthesia, Royal Victoria Hospital, McGill University Health Center, Montréal, QC, Canada
[3] Center for Clinical Epidemiology, Assistance Publique Hôpitaux de Paris, Hotel Dieu Hospital, University Paris Descartes, Paris, France
[4] Intensive Care Unit, Instituto Estadual do Cérebro Paulo Niemeyer, Rio de Janeiro, Brazil
[5] Instituto Nacional de Infectologia Evandro Chagas, Fundação Oswaldo Cruz (Fiocruz), Rio de Janeiro, Brazil
[6] Intensive Care Unit, Hospital das Américas, Rio de Janeiro, Brazil
[7] Department of Anesthesia, Critical Care and Emergency, Spedali Civili University Hospital, Brescia, Italy
[8] Department of Medical and Surgical Specialties, Radiological Sciences and Public Health, University of Brescia, Brescia, Italy
[9] General Intensive Care Unit, Assistance Publique Hôpitaux de Paris, Raymond-Poincaré Hospital, University of Versailles Saint-Quentin en Yvelines, Garches, France
[10] Department of Physiology, INSERM U 1179, Assistance Publique Hôpitaux de Paris, Raymond-Poincaré Hospital, University of Versailles Saint-Quentin en Yvelines, Garches, France
[11] Laboratory of Human Histopathology and Animal Models, Institut Pasteur, Paris, France
[12] D'Or Institute for Research and Education, Rio de Janeiro, Brazil
[13] Department of Neurology, Intensive Care Unit, Groupe Hospitalier Pitié-Salpêtrière, AP-HP, Paris, Sorbonne Universités, Faculté de Médecine Pitié-Salpêtrière, Paris, France, Paris, France
[14] Department of Neurology, Critical Care Neurology, Columbia University, New York, NY, USA
[15] Neuro-Anesthesiology and Intensive Care Unit, Sainte-Anne Teaching Hospital, University of Paris-Descartes, Paris, France

Corresponding authors
Benjamin Rohaut,
br2529@columbia.edu
Tarek Sharshar,
t.sharshar@ch-sainte-anne.fr

## ABSTRACT

**Background:** Deep sedation may hamper the detection of neurological deterioration in brain-injured patients. Impaired brainstem reflexes within the first 24 h of deep sedation are associated with increased mortality in non-brain-injured patients. Our objective was to confirm this association in brain-injured patients.
**Methods:** This was an observational prospective multicenter cohort study involving four neuro-intensive care units. We included acute brain-injured patients requiring deep sedation, defined by a Richmond Assessment Sedation Scale (RASS) < −3. Neurological assessment was performed at day 1 and included pupillary diameter, pupillary light, corneal and cough reflexes, and grimace and motor response to noxious stimuli. Pre-sedation Glasgow Coma Scale (GCS) and Simplified Acute

Physiology Score (SAPS-II) were collected, as well as the cause of death in the Intensive Care Unit (ICU).

**Results:** A total of 137 brain-injured patients were recruited, including 70 (51%) traumatic brain-injured patients, 40 (29%) vascular (subarachnoid hemorrhage or intracerebral hemorrhage). Thirty patients (22%) died in the ICU. At day 1, the corneal (OR 2.69, $p = 0.034$) and cough reflexes (OR 5.12, $p = 0.0003$) were more frequently abolished in patients that died in the ICU. In a multivariate analysis, abolished cough reflex was associated with ICU mortality after adjustment to pre-sedation GCS, SAPS-II, RASS (OR: 5.19, 95% CI [1.92–14.1], $p = 0.001$) or dose of sedatives (OR: 8.89, 95% CI [2.64–30.0], $p = 0.0004$).

**Conclusion:** Early (day 1) cough reflex abolition is an independent predictor of mortality in deeply sedated brain-injured patients. Abolished cough reflex likely reflects a brainstem dysfunction that might result from the combination of primary and secondary neuro-inflammatory cerebral insults revealed and/or worsened by sedation.

## INTRODUCTION

Patients with severe brain injury frequently receive early deep sedation (*Oddo et al., 2016*). Deep sedation may improve cerebral hemodynamics by reducing the rate of cerebral oxygen consumption and by decreasing intracranial pressure (ICP). However, deep sedation may also delay awakening, induce delirium (*Oddo et al., 2016*) and, increase mortality (*Shehabi et al., 2018*). In addition, sedation hampers clinical assessment of neurological status in brain-injured patients, potentially masking acute neurological worsening. Deep sedation also compromises the assessment of the patient's prognosis, which then relies on the pre-sedation examination (such as Glasgow Coma Scale (GCS)). It is therefore challenging for ICU-physicians to routinely assess the neurological status of severely brain-injured patients requiring deep sedation. Nonetheless, ICU physicians have at their disposal various tools, including neurological examination and electrophysiological testing. The main difficulty, then, is to distinguish the effects of sedation from the consequences of underlying brain injury on clinical signs and the activity of the electroencephalogram (EEG). This is a highly complex situation compounded by the existence of both primary and secondary brain insults. Clinical relevance of neurological examination—especially brainstem reflex assessment—has previously been demonstrated in non-brain-injured, critically-ill patients (*Foo, Loan & Brennan, 2019*), including deeply sedated patients (*Sharshar et al., 2011a*; *Rohaut et al., 2017*; *Azabou et al., 2017, 2018*). Assessment of brainstem reflexes is feasible and reproducible, and constitutes an early independent predictor of ICU-mortality, after adjustment to critical illness severity, sedation level, and sedative doses in non-brain-injured, critically ill patients

(*Sharshar et al., 2011a*; *Rohaut et al., 2017*; *Azabou et al., 2017, 2018*). Our pathophysiological hypothesis is that critical illness may be associated with brainstem dysfunction, which might itself be caused by the combined effects of critical illness and sedation, and may contribute to mortality, notably via a central autonomic dysfunction (*Sharshar et al., 2011a*; *Rohaut et al., 2017*; *Benghanem et al., 2020*).

A previous study has shown the prognostic value of early (day 1) assessment of brainstem reflexes in deeply sedated, non-brain-injured patients (*Sharshar et al., 2011a*). The main objective of the present study was to extend these previous findings to brain-injured patients.

## MATERIALS AND METHODS

### Study design and setting

This was a prospective, multicenter, international observational cohort study, approved by the ethics committee of Paris (Ile de France IV), France (Approval number 2014-AO1102-45) and Brescia (NP 1840, 04-11-2014), Italy. Written informed consent was obtained from the patients' legal representative in France. In Italy, the Ethics Committee waived the requirement for consent because relatives are not regarded as legal representatives of the patient in the absence of a formal designation. The current study was a pilot study preceding the design of a larger prospective multicentre study assessing the prognostic value of brainstem dysfunction in sedated critically ill patients (ClinicalTrials.gov number: NCT02395861). Patients were recruited in four intensive care units, including one neuro ICU and three medical and surgical ICUs. Patients were recruited between December 2011 and February 2015. The Strengthening the Reporting Studies in Epidemiology (STROBE) guidelines were followed thoroughly (*Von Elm et al., 2007*).

### Participants

Adult patients were eligible if they were deeply sedated following a major brain injury: severe traumatic brain injury, subarachnoid or intraparenchymal cerebral hemorrhage, ischemic stroke, or following a complicated neurosurgical or endovascular procedure. Deep sedation was defined by a Richmond Assessment Sedation Scale (RASS) score < −3 (*Ely et al., 2003*). Patients were included if sedation lasted between 12 and 24 h, and could be neurologically assessed by one of the PI (SK, JA, CR, CV, FR, NH, GM, GS, NL and FB) at working hours.

Patients were excluded if they were affected by a peripheral neurologic disorder involving the cranial nerves (e.g., Guillain Barré syndrome), or had been referred following cardiac arrest. Based on our previous findings in non-brain-injured patients, we planned to include 150 patients in the present study (*Sharshar et al., 2011a*; *Rohaut et al., 2017*).

### Sedation

Decisions to initiate or to withdraw sedation as well as the level of sedation were made by the physician in charge of the patient. We recorded the time and reason for initiating deep sedation as well as the GCS prior to sedation. Since sedation was administered as part of the treatment of the cerebral insult, no systematic interruptions of sedation or

decreasing trials were performed during the first 24 h. Nevertheless, depth of sedation was monitored using the RASS every 4 h. After the first 24 h, the possibility of discontinuing sedation was assessed on a daily basis. Titration of sedation was performed at least twice daily, targeting physician in charge-defined RASS levels. The date of awakening, defined by spontaneous eye opening and visual contact >10 s (i.e., RASS ≥ −1), was recorded.

Sedation was obtained through a continuous infusion of midazolam and/or propofol, in combination with sufentanil or fentanyl. Total cumulative doses of administered drugs at the time of neurologic examination were collected.

## Neurologic examination

The detailed procedure pertaining to neurological examination has been previously described (Sharshar et al., 2011a). Briefly, we assessed: (1) depth of sedation (RASS); (2) reactivity, using the motor and eye response components of the GCS; (3) brainstem reflexes, including pupil size (miosis, normal, or mydriasis), pupillary light reflex, corneal reflex, facial muscle movement in response to noxious stimulation of the temporo-mandibular joint, and the cough reflex in response to tracheal suctioning.

Neurologic examination was performed after 24 ± 12 h of continuous sedation (day 1). Retaining the same definition used in our previous studies, corneal and pupillary reflexes were considered abolished only when both right and left side reflex were abolished. The oculocephalic response to lateral passive head rotation was not performed in traumatic brain injured patients. The Full Outline of Unresponsiveness (FOUR) score was recorded (Wijdicks et al., 2005). At the time of neurological examination, ICP was recorded if available. Intracranial hypertension was defined as ICP > 25 mmHg lasting for more than 5 min or by the need of an additional treatment to control the ICP (additional sedative agent, ventricular drainage, craniotomy with hematoma evacuation or decompressive craniectomy).

## Baseline clinical and biological and imaging data

Demographic characteristics, body weight, date, time and cause of ICU admission, co-morbidities using McCabe score (McCabe & Jackson, 1962), date of invasive mechanical ventilation (MV) initiation and its duration, ICU length of stay, occurrence of microbiologically documented ventilator-associated pneumonia as well as the date and cause of death were recorded. The Simplified Acute Physiological Score II (SAPS-II) (Le Gall et al., 2005), the Sequential Organ Failure Assessment (SOFA) (Vincent et al., 1996) as well as key interventions and standard biological tests needed to calculate these scores were recorded. Clinical, biological and neuroradiological data were collected as part of the routine care.

The presence or absence of infratentorial lesions, including brainstem lesions, was assessed on the first cerebral computed tomography scanner (CT-scan) performed within 24 h following ICU admission, by the senior neuroradiologist and ICU physician in charge.

## Outcomes

Primary recorded outcome was ICU mortality. Secondary outcomes were the occurrence of delayed awakening and of delirium following sedation discontinuation. Delayed awakening was defined by RASS < −1 despite discontinuation of sedation for more than 72 h. Delirium was assessed daily using the confusion assessment method for the ICU (CAM-ICU) (*Ely et al., 2001*). Other secondary outcomes were the duration of MV, the length of stay in the ICU and the occurrence of microbiologically documented ventilator-associated pneumonia. Patients had follow-up until ICU discharge.

Withdrawal of life-sustaining therapies (WLST) was determined according to applicable French and Italian Law. In current practice, no such decision involves the result of early neurological examination in a deeply-sedated patient.

## Bias and confounding factors

We sought to mitigate potential confounding factors that might influence neurological examination as well as relevant outcomes, namely mortality and the occurrence of delirium. Neurological examination was performed by senior ICU physicians who were either neurologists or neuro-intensivists, or were specifically trained by a senior neurologist. The investigator was different from the clinician in charge of the patient. Inter-observer agreement for such an examination has been shown to be satisfactory (kappa scores ranged from 0.62 to 1) (*Sharshar et al., 2011b*). Brainstem reflexes are routinely assessed in neuro-ICU patients. Management of deep sedation was assessed by collecting cumulative doses and duration of sedation as well as daily RASS. The cause of death and its main risk factors were also assessed, including the GCS prior to sedation, SAPS-II and the SOFA score as well as the cause of brain injury.

## Statistical analysis

Data are reported as numbers (percentage), mean (standard deviation), or median (inter-quartile range). Groups were compared using the Wilcoxon rank sum test or the chi-square test. Logistic regression was used to explore the associations between the pre-sedation GCS, SAPS-II, RASS scores, sedation doses, cough reflex, corneal reflex, and ICU mortality. These variables were determined a priori according to our previous findings (*Rohaut et al., 2017*). Since the number of events was limited, we used Firth penalized logistic regression for multivariable models, in order to limit small-sample bias. Discrimination of multivariable models was assessed using the concordance ($c$) index, which is equivalent to the area under the receiver operating characteristics curve. It varies theoretically between 0.5 and 1.0, a value of 1.0 indicating a perfect discrimination. Missing data were handled by multiple imputation by chained equations, all variables considered for analysis being used in the imputation model. Since about half patients had at least one variable missing (35 patients had missing data for one variable, one had missing data for seven), 50 imputed datasets were generated (*White, Royston & Wood, 2011*). Each imputed dataset was analyzed separately, and estimates were then pooled to provide point estimates and confidence intervals (CI) (*Marshall et al., 2009*). $P$-values < 0.05 were considered as statistically significant. All analyses were carried out

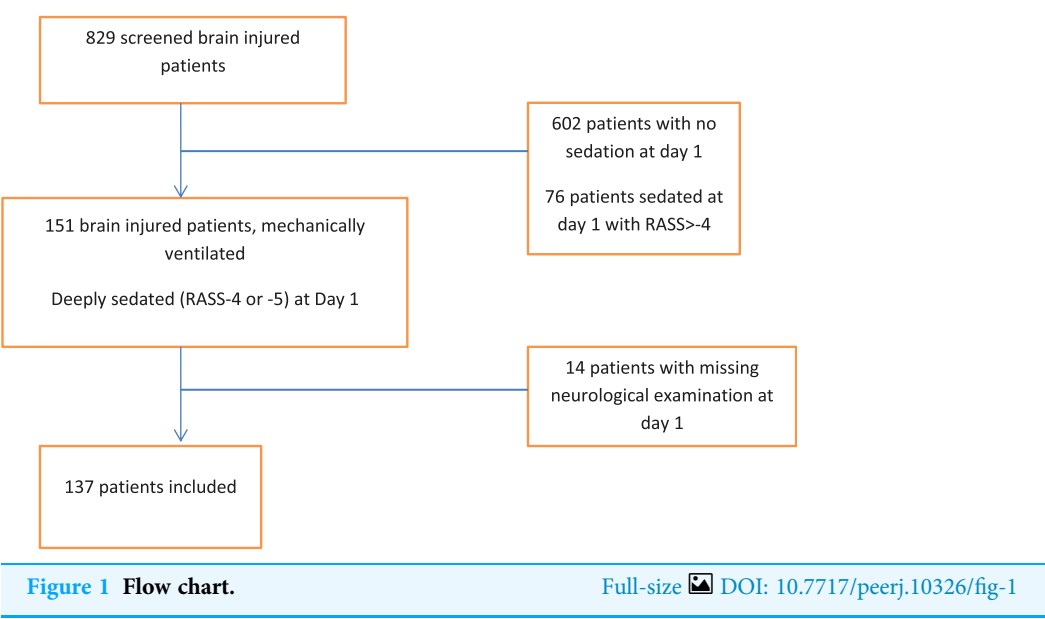

Figure 1 Flow chart.               

using the R statistical software version 3.4.1 (The R Foundation for Statistical Computing, Vienna, Austria).

## RESULTS

### Patient characteristics

Among 151 consecutive brain injured patients receiving deep sedation within 12 h of admission, a total of 14 were excluded (Fig. 1). Overall, 137 patients were enrolled; their baseline characteristics are presented in Table 1. Cause of brain injury was blunt trauma in 70 (51%) patients, vascular in 40 patients (29%) and miscellaneous (mainly post-surgical or endovascular procedure) in 27 (20%). MV was initiated for neurological reasons (airway management in comatose patients) in 120 (88%) patients. Other reasons for intubation and MV were acute respiratory failure in 6 patients (4%), shock in one patient (1%), post-operative period in 6 patients (4%), and undetermined in 4 patients (3%). The median age was 50 (34–63) years, 80 (59%) patients were male, and the median SAPS II was 46 (36–55).

### Neurological features of the patients

Main neurological features are presented in Table 2. The median GCS prior to sedation was 7 (4–11). At the time of inclusion, all patients were deeply sedated (RASS < −3), with 90 (66%) patients exhibiting RASS-5. Deep sedation was obtained using one ($n = 114$, 83%), two ($n = 21$, 15%) or three ($n = 2$, 2%) hypnotic agents. One hundred twenty-two (89%) patients received midazolam, 35 (26%) propofol and 5 (4%) sodium thiopental. Sufentanil was administered to 107 (78%) patients. At inclusion, median FOUR score was 4 (2–5). The most frequently preserved brainstem reflex was the corneal reflex, present in 103 patients (79%), and grimacing to pain was the most frequently abolished reflex (present in 32 patients, 30%). The cough reflex was present in 100 patients (74%). The proportion of patients with RASS-5 was greater in patients with abolished cough reflex

**Table 1 Characteristics of patients and ICU outcome.**

| Variable | N | All patients | TBI | Vascular | Other |
|---|---|---|---|---|---|
| No. patients (% of total) | | 137 | 70 (51) | 40 (29) | 27 (20) |
| Age (years)—median (IQR) | 137 | 50 (34 to 63) | 39 (25 to 53) | 60 (52 to 66) | 53 (40 to 77) |
| Female—no. (%) | 137 | 56 (41) | 16 (23) | 27 (68) | 13 (50) |
| SAPS-II—median (IQR) | 129 | 46 (36 to 55) | 41 (33 to 53) | 48 (38 to 52) | 51 (44 to 60) |
| Initiation of mechanical ventilation—no. (%) | 137 | | | | |
| Coma | | 120 (88) | 68 (97) | 34 (85) | 18 (67) |
| Acute respiratory failure | | 6 (4) | 0 (0) | 2 (5) | 4 (15) |
| Shock | | 1 (1) | 0 (0) | 0 (0) | 1 (4) |
| Surgery/procedure | | 6 (4) | 0 (0) | 4 (10) | 2 (7) |
| Other | | 4 (3) | 2 (3) | 0 (0) | 2 (7) |
| McCabe score 1—no. (%) | 135 | 106 (79) | 64 (93) | 22 (55) | 20 (77) |
| GCS at admission—median (IQR) | 132 | 7 (4 to 10) | 7 (5 to 10) | 7 (4 to 10) | 6 (3 to 11) |
| SOFA at admission—median (IQR) | 132 | 6 (4 to 8) | 5 (4 to 8) | 5 (4 to 7) | 7 (4 to 9) |
| Outcome | | | | | |
| Duration of mechanical Ventilation (days)—median (IQR) | 131 | 13 (8 to 24) | 14 (10 to 23) | 13 (8 to 20) | 9 (5 to 23) |
| Duration of sedative infusion (days)—median (IQR) | 133 | 5 (3 to 10) | 5 (3 to 12) | 4 (3 to 11) | 4 (3 to 7) |
| Occurrence of elevated intracranial pressure—no. (%) | 103 | 50 (49) | 31 (48) | 16 (50) | 3 (43) |
| Occurrence of ventilator acquired pneumonia—no. (%) | 135 | 86 (64) | 50 (72) | 25 (62) | 11 (42) |
| Time to awakening (days)—median (IQR) | 131* | 4 (1 to 12) | 4 (1 to 12) | 6 (2 to 17) | 1 (0 to 5) |
| Occurrence of delayed awakening—no. (%) | 131* | 67 (51) | 36 (51) | 23 (64) | 8 (32) |
| Occurrence of delirium—no. (%) | 112† | 66 (59) | 46 (75) | 12 (40) | 8 (38) |
| Duration of ICU stay (days)—median (IQR) | 136 | 19 (11 to 32) | 19 (11 to 32) | 20 (12 to 36) | 15 (8 to 24) |
| Death in ICU—no. (%) | 137 | 30 (22) | 12 (17) | 11 (28) | 7 (26) |

Notes:

*N*, number of evaluated patients; SAPS II, Simplified Acute Physiology Score; GCS, Glasgow Coma Scale; SOFA, sepsis related organ failure assessment; ICU, Intensive Care Unit; TBI, Traumatic brain injury; IQR, Interquartile range (Q1 to Q3).

* Patients who died while sedated could not be evaluated.

† Patients who died while sedation or before awakening could not be evaluated.

(80% vs. 60%, $p = 0.039$). Similarly, the cumulative doses administrated at time of clinical exam of midazolam and opioid agents were greater in patients with abolished cough reflex (1.2 vs. 1.9 mg/Kg, $p = 0.005$ and, 5.2 vs. 8.5 µg/Kg, $p = 0.009$ respectively) (Table 3).

Brain imaging was performed for 133 patients (97%) by a CT scan, of whom 45 (34%) patients exhibited infra-tentorial lesions including brain stem lesions.

## ICU mortality

Thirty patients (22%) died in the ICU (Table 1). Causes of death were multi-organ failure in 6 patients (20%), brain death in 5 patients (17%) and, withdrawal of life sustaining therapies (WLST) in 16 patients (53%). Cause of death was not reported for 3 patients (10%). WLST-related death occurred at day 8 (6–17) (ranging from day 3 to day 48 after admission). SAPSII (OR 1.06 per unit, $p = 0.001$) was significantly greater in non-survivors (Table 4). At admission, GCS were significantly lower (OR 0.80 per unit, $p = 0.004$) in non-survivors. The proportion of patients with RASS-5 ($p = 0.32$) did not statistically differ between the two groups. The corneal (OR 2.69, $p = 0.034$) and cough reflexes

Table 2 **Neurological assessment at inclusion.**

| Variable | N | All patients | TBI | Vascular | Other |
|---|---|---|---|---|---|
| No. patients (% of total) | | 137 | 70 (51) | 40 (29) | 27 (20) |
| RASS-5—no. (%) | 137 | 90 (66) | 54 (77) | 21 (52) | 15 (56) |
| Neurologic response | | | | | |
| GCS motor—median (IQR) | 124 | 1 (1 to 1) | 1 (1 to 1) | 1 (1 to 2) | 1 (1 to 3) |
| GCS ocular—median (IQR) | 124 | 1 (1 to 1) | 1 (1 to 1) | 1 (1 to 1) | 1 (1 to 1) |
| FOUR Score—median (IQR) | 111 | 4 (2 to 5) | 4 (2 to 5) | 4 (2 to 5) | 4 (2 to 7) |
| Eye response—median (IQR) | 113 | 0 (0 to 0) | 0 (0 to 0) | 0 (0 to 0) | 0 (0 to 0) |
| Motor response—median (IQR) | 113 | 0 (0 to 0) | 0 (0 to 0) | 0 (0 to 0) | 0 (0 to 2) |
| Brainstem reflexes—median (IQR) | 111 | 4 (2 to 4) | 4 (2 to 4) | 4 (2 to 4) | 3 (2 to 4) |
| Respiration—median (IQR) | 113 | 0 (0 to 1) | 0 (0 to 1) | 0 (0 to 1) | 1 (0 to 1) |
| Brainstem reflexes impairment | | | | | |
| Pupillary light reflex—no. (%) | 134 | 55 (41) | 29 (42) | 19 (49) | 7 (27) |
| Corneal reflex—no. (%) | 131 | 28 (21) | 15 (23) | 7 (18) | 6 (23) |
| Grimacing to pain—no. (%) | 108 | 76 (70) | 51 (86) | 14 (48) | 11 (55) |
| Cough reflex—no. (%) | 135 | 35 (16) | 19 (28) | 9 (22) | 7 (27) |
| Myosis—no. (%) | 136 | 69 (51) | 37 (54) | 19 (48) | 13 (48) |

Note:
N, number of evaluated patients; IQR, Interquartile range (Q1 to Q3); RASS, Richmond Assessment Sedation Scale; GCS, Glasgow Coma Scale; FOUR score, Full outline of Unresponsiveness; TBI, Traumatic brain injury.

Table 3 **Comparison of sedation according to the presence/absence of cough reflex.**

| | Cough reflex | | P |
|---|---|---|---|
| | Present | Absent | |
| No. patients | 100 | 35 | |
| Hypnotic agent (several possible) | | | |
| Midazolam—no. (%) | 86 (86) | 34 (97) | 0.11 |
| Propofol—no. (%) | 25 (25) | 10 (29) | 0.66 |
| Thiopental—no. (%) | 1 (1) | 4 (11) | 0.016 |
| Midazolam dose (mg/Kg)—median (IQR) | 1.2 (0.7 to 2.1) | 1.9 (1.2 to 2.3) | 0.005 |
| Propofol dose (mg/Kg)—median (IQR) | 17.7 (10.2 to 22.6) | 37.1 (18.4 to 44.4) | 0.12 |
| Morphinic agent—no. (%) | | | 0.60 |
| No | 4 (4) | 0 (0) | |
| Sufentanyl | 78 (78) | 27 (77) | |
| Fentanyl | 18 (18) | 8 (23) | |
| Morphinic dose (µg/Kg)*—median (IQR) | 3.9 (2.2 to 7.4) | 6.7 (3.8 to 10.3) | 0.006 |
| SAPS-II—median (IQR) | 46 (36 to 55) | 46 (34 to 55) | 0.76 |
| SOFA at admission—median (IQR) | 5 (4 to 8) | 6 (5 to 8) | 0.27 |
| RASS-5—no. (%) | 60 (60) | 28 (80) | 0.039 |

Notes:
IQR, Interquartile range (Q1 to Q3); Sedation and analgesics doses are cumulative doses from introduction to neurological assessment (12–24 h); two patients were excluded from this analysis, for missing sedation data.
* Sufentanyl-equivalent dose (for the 18% of patients which received fentanyl, doses where divided by 10).

**Table 4 Association of neurological response with ICU death.**

| | Unadjusted | | Multiple model 1 | | Multiple model 2 | |
|---|---|---|---|---|---|---|
| | OR (95% CI) | *P* | aOR (95% CI) | *P* | aOR (95% CI) | *P* |
| SAPS-II (per unit) | 1.06 [1.02–1.09] | 0.002 | 1.04 [1.00–1.08] | 0.086 | 1.04 [0.99–1.08] | 0.092 |
| GCS at admission (per unit) | 0.80 [0.69–0.93] | 0.004 | 0.88 [0.74–1.04] | 0.13 | 0.87 [0.72–1.04] | 0.12 |
| RASS-5 | 1.58 [0.64–3.91] | 0.32 | 0.74 [0.26–2.17] | 0.59 | – | – |
| Midazolam dose (mg/Kg) | 0.69 [0.44–1.11] | 0.12 | – | – | 0.42 [0.14–1.25] | 0.12 |
| Morphinic dose (µg/Kg)* | 0.97 [0.88–1.08] | 0.59 | – | – | 1.01 [0.82–1.25] | 0.91 |
| Absent cough reflex | 5.12 [2.13–12.4] | 0.0003 | 5.19 [1.92–14.1] | 0.001 | 8.89 [2.64–30.0] | 0.0004 |
| Absent corneal reflex | 2.69 [1.08–6.68] | 0.034 | 1.71 [0.57–5.10] | 0.34 | 1.66 [0.54–5.08] | 0.38 |
| *c* index (95% CI) | | | 0.810 [0.726–0.893] | | 0.852 [0.773–0.931] | |

**Notes:**
* Sufentanyl-equivalent dose.
OR, odds ratio; aOR, adjusted odds ratio; SAPS II, Simplified Acute Physiology Score; RASS, Richmond Assessment Sedation Scale.

(OR 5.12, $p = 0.0003$) were more frequently abolished in non-survivors. After adjustment to pre-sedation GCS, SAPS-II, and RASS (OR: 5.19, 95% CI [1.92–14.1], $p = 0.001$) or midazolam and sufentanyl doses (OR: 8.89, 95% CI [2.64–30.0], $p = 0.0004$), an abolished cough reflex was associated with ICU mortality.

## Other outcomes

Median duration of sedation was 5 (3–10) days while median duration of MV was 13 (8–24) days. Median time to awakening was 4 (1–12) days following discontinuation of sedation. Delayed awakening was observed in 67 patients (51%) and delirium in 66 patients (59%). Eighty-six patients (64%) developed ventilator associated pneumonia. Among the 103 patients in whom ICP was monitored, 50 patients (49%) suffered from at least one episode of elevated ICP.

## DISCUSSION

In this prospective, multicenter cohort study we found that, in brain injured patients requiring deep sedation, early abolition of the cough reflex (day 1) was associated with ICU mortality after adjustment for severity of illness (i.e., SAPS-II), brain injury (i.e., pre-sedation GCS), depth of sedation (i.e., RASS) and sedative and analgesic doses. Moreover, abolition of the cough reflex was independent of the type of injury.

These results extend our previous findings obtained in non-brain injured patients (*Sharshar et al., 2011a*; *Rohaut et al., 2017*). It supports our hypothesis that mortality could result from dysfunction of the brainstem, which controls vital functions via the autonomic nervous system (*Heming et al., 2017*). Thus, abolition of the cough reflex could be a clinical marker of dysfunction in the medulla, which integrates cardiovascular and respiratory centers. Interestingly, critical illness is associated with a reduction in heart rate and tidal volume variability, which, as markers of impaired central control, are associated with organ failure, mortality, and weaning failure from MV (*Annane et al., 1999*; *Wysocki et al., 2006*).

The putative mechanisms of brainstem dysfunction can be related to three non–mutually exclusive mechanisms: (1) primary brain injury; (2) neuro-inflammation; (3) oversedation (*Benghanem et al., 2020*).

1. Cough reflex abolition may result from a direct brainstem injury related to hemorrhage, axonal lesions, herniation brain swelling etc. Specific assessment of brainstem injury would have required specific brain imaging (e.g., magnetic resonance imaging, diffusion tensor imaging) (*Fischer et al., 2016*) which was out of the scope of this observational study. Patient management, including the choice of brain imaging within the first 24 h, was under the responsibility of ICU-physicians. In this context, CT scan is the most commonly performed exam (*Connolly et al., 2012*; *Geeraerts et al., 2018*).

2. In addition to the direct injury, systemic inflammation can cause neuro-inflammatory brainstem injuries. These lesions are usually radiologically undetectable. Circulating mediators can directly reach the brainstem through the area postrema. This zone's lack of a blood-brain barrier allows neuro-inflammatory insults and neuronal apoptosis, notably within the medullary autonomic and respiratory centers (*Sharshar et al., 2002*, *2003*). As described in septic acute encephalopathy or delirium in ICU, this mechanism could occur as early as 24 h after brain injury (*Mazeraud, Bozza & Sharshar, 2018*; *Slooter et al., 2020*).

3. Finally, as respiratory centers are sensitive to sedation (*Rohaut et al., 2017*) especially opioids, cough reflex abolition could be a direct effect of sedation. The dose of sedatives and opioids and the depth of sedation (according to the RASS) were greater in patients with absent cough reflex, which can therefore be considered a marker of oversedation. However, abolition of cough reflex remains associated with mortality after adjustment on sedation levels as well as sedative and analgesic doses, suggesting that its abolition integrates other processes than sedation alone. It would be interesting to test the predictive value of cough reflex abolition against EEG, which may be more accurate to assess depth of sedation.

The conclusion of our study is pragmatic; severity and prognostic assessment of brain-injured patients requiring deep sedation should integrate assessment of brainstem reflexes, especially the cough reflex. This assessment can be performed by using the FOUR score (*Wijdicks et al., 2005*; *Foo, Loan & Brennan, 2019*). It must be noted that information regarding the neurological components of the SAPS-II and SOFA severity scores, as well as of the RASS sedation scale, is limited. Indeed, the SAPS-II and SOFA score include the GCS value before sedation; while the RASS relies solely on the patient's motor reactivity to verbal or non-painful physical stimulation. Brainstem reflex assessment within the first 24 h of sedation provides clinicians with a comprehensive and temporally integrative neurological evaluation.

## Limits

Our study has several limits. First, the causes of brain injuries are multiple. Nevertheless, except for grimacing to pain and pupillary light reflexes, the neurological patterns were

similar among the different subpopulations (Table 2). Our results suggest that the association between cough reflex abolition and ICU mortality is independent from the cause of brain injury. Second, the optimal way to assess the depth of sedation remains a matter of debate. In accordance with our previous findings, we chose to assess the depth of sedation using a specifically designed clinical scale (the RASS; Table 4, model 1). However, since doses of sedation were significantly different within patients with preserved and abolished cough reflex, we also decided to include this parameter in our multivariate analysis (Table 4, model 2). Continuous EEG monitoring and/or determining plasma levels of sedatives could have provided more detail regarding the level of sedation, however these assessments are not routinely undertaken. Based on our results, and without measurement of plasmatic sedation levels, assessing the cause of cough reflex inhibition is not feasible. It would be interesting to determine the prognostic value of the cough reflex against EEG, and to assess whether its abolition depends on circulating levels of sedatives. Third, as the main cause of death was WLST, our study could have been exposed to the self-fulfilling prophecy bias. However, the proportion of WLST in our study was comparable to other reports (Leblanc et al., 2018). In addition, it is not a common clinical practice to base WLST solely on day 1 neurological examination. In our study, no WLST-related death occurred before day 3. Nevertheless, a neurological examination at day 1, 2, and 3 could have been more valuable than solely at day 1, and this must be further evaluated. It must also be noted that since the reason for ICU discharge was not prospectively assessed, we cannot rule out that patients have been discharged from ICU for palliation on the ward and have died on the ward. Only ICU mortality at day 28 was addressed. Fourth, our data does not allow any exploration of the causality of brainstem dysfunction. In addition to the proposed mechanisms (i.e., primary injury, oversedation or neuroinflammation), the causal link between brainstem failure and multi-organ failure (which was responsible for 20% of the mortality in our study) has to be explored. Understanding to what extend brainstem dysfunction could be a cause or a consequence of multi-organ failure would need further investigation.

## CONCLUSIONS

Taken together, our results suggest that an absence of cough reflex is an independent predictor of ICU-mortality in deeply sedated brain-injured patients. This is an important extension of our previous findings in deeply sedated non-brain-injured ICU patients. Abolition of cough reflex possibly reflects a brainstem dysfunction that could compromise vital functions. This brainstem dysfunction might result from over-sedation combined with primary and secondary cerebral insults. Assessment of brainstem reflexes may complete SAPS-II and RASS scores for evaluating the severity and depth of sedation, and help predict the prognosis of deeply sedated brain injured patients. Future research could aim at assessing the safety and efficacy of sedation titration aimed at preserving the cough reflex without compromising the control of ICP in brain-injured patients.

## ACKNOWLEDGEMENTS

This work is dedicated to the memory of Professor Jean Mantz. We would like to thank all the members of the "Groupe d'Explorations Neurologiques en Réanimation (GENER)", leaded by Professor SHARSHAR (email: t.sharshar@ch-sainte-anne.fr), namely: Nicolas ADAM, Jeremy ALLARY, Eric AZABOU, Omar BEN HADJ SALEM, Francis BOLGERT, Antoine BRACONNIER, Alain CARIOU, Patrick CHILLET, Fabrice CHRETIEN, Vincent DEGOS, Sophie DEMERET, Raphael GAILLARD, Stephane GAUDRY, Nicholas HEMING, Tarik HISSEM, Stanislas KANDELMAN, Jonathan LEVY, Yann L'HERMITTE, Eric MAGALHAES, Jean MANTZ, Aurélien MAZERAUD, Régine MORIZOT-KOUTLIDIS, Lionel NACCACHE, Vincent NAVARRO, Hervé OUTIN, Andrea POLITO, Benjamin ROHAUT, Tarek SHARSHAR, Shidasp SIAMI, Romain SONNEVILLE, Pierre ROCHETEAU, Franck VERDONK, Chuyen VU DINH and Nicolas WEISS.

### Funding

The authors received no funding for this work.

### Competing Interests

The authors declare that they have no competing interests.

### Author Contributions

- Stanislas Kandelman conceived and designed the experiments, performed the experiments, analyzed the data, prepared figures and/or tables, authored or reviewed drafts of the paper, and approved the final draft.
- Jérémy Allary conceived and designed the experiments, performed the experiments, analyzed the data, prepared figures and/or tables, and approved the final draft.
- Raphael Porcher conceived and designed the experiments, analyzed the data, prepared figures and/or tables, and approved the final draft.
- Cássia Righy performed the experiments, authored or reviewed drafts of the paper, and approved the final draft.
- Clarissa Francisca Valdez performed the experiments, authored or reviewed drafts of the paper, and approved the final draft.
- Frank Rasulo performed the experiments, authored or reviewed drafts of the paper, and approved the final draft.
- Nicholas Heming performed the experiments, authored or reviewed drafts of the paper, and approved the final draft.
- Guy Moneger performed the experiments, authored or reviewed drafts of the paper, and approved the final draft.
- Eric Azabou performed the experiments, authored or reviewed drafts of the paper, and approved the final draft.

- Guillaume Savary performed the experiments, authored or reviewed drafts of the paper, and approved the final draft.
- Djillali Annane conceived and designed the experiments, authored or reviewed drafts of the paper, and approved the final draft.
- Fabrice Chretien conceived and designed the experiments, authored or reviewed drafts of the paper, and approved the final draft.
- Nicola Latronico performed the experiments, authored or reviewed drafts of the paper, and approved the final draft.
- Fernando Augusto Bozza performed the experiments, authored or reviewed drafts of the paper, and approved the final draft.
- Benjamin Rohaut conceived and designed the experiments, analyzed the data, prepared figures and/or tables, authored or reviewed drafts of the paper, and approved the final draft.
- Tarek Sharshar conceived and designed the experiments, analyzed the data, authored or reviewed drafts of the paper, and approved the final draft.

### Human Ethics

The following information was supplied relating to ethical approvals (i.e., approving body and any reference numbers):

Ethics committee of Paris (Ile de France IV), France (Approval number 2014-AO1102-45) and Brescia (NP 1840, 04-11-2014), Italy.

The current study was a pilot study preceding the design of a larger prospective multicentre study assessing the prognostic value of brainstem dysfunction in sedated critically ill patients (ClinicalTrials.gov number: NCT02395861).

### Data Availability

The raw data is available in the Supplemental File.

### Supplemental Information

Supplemental information for this article can be found online at http://dx.doi.org/10.7717/peerj.10326#supplemental-information.

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
