# Peer review of "Early abolition of cough reflex predicts mortality in deeply sedated brain-injured patients"

_PeerJ, doi:10.7717/peerj.10326_

## Round 0.1 · original submission · Major Revisions

Dear Authors,

Please revise your manuscript according to the comments of the two reviewers.

·

Basic reporting

- I would like to congratulate the authors for the interesting study. I commend the use of the STROBE reporting guidelines. I think the study design (cohort study) should be mentioned in the title or abstract and the study design section.

- Follow-up / loss to follow-up have not been mentioned but presumably every participant had follow-up until they were discharged from ICU?

- It has been mentioned in text that FOUR score was lower and SOFA and age were higher in non-survivors. I think Table 4 should include this data for clarity.

- The p value for GCS (line 257) does not match the value in Table 4.

Experimental design

- Recruitment method - were all patients admitted to the designated ICU recruited and assessed for eligibility or the investigators decided which patients to be recruited?

- Study size - how was the size determined? Was power calculation done?

- The efforts to mitigate potential confounding factors is laudable.

Validity of the findings

- I think the limitations of the study have been addressed adequately at the discussion.

- The conclusion states that early cough reflex abolition is an independent predictor of mortality in deeply sedated brain-injured patients. Perhaps it is more accurate to limit that to ICU mortality as patients did not get follow-up after they were discharged from ICU? Had there been any patients discharged from ICU for palliation on the ward and had these been included in the mortality count?

- As you have mentioned in the limitations, doses of sedation were significantly different between those with preserved and abolished cough reflex. 20% of the ICU mortality died secondary to multi-organ failure. To what extent do you think the absence of cough reflex is a result of the high levels of sedative when the plasma level is unknown, and whether this might change the conclusion?

Additional comments

- Some studies have shown that neurological assessment within the first day of brain injury may not be reliable / as good as those performed later in terms of predicting mortality. As you mentioned, it is not common practice to base decision for withdrawal of life supporting treatment solely on day 1 neurological examination findings. Would it be more valuable to assess the prognostication of cough reflex later (e.g. day 3)?

·

Basic reporting

Grammar and style are, by and large, acceptable but could be improved by having a native speaker read through the manuscript.

Scientific background and context, text structure and literature references are adequate. Results are self-contained and relevant to the main hypothesis about brainstem dysfunction in the ICU.

Experimental design

This is an observational study with research questions that are within the aims and scope of the journal. Ethical and technical standards are explained in sufficient detail.

Validity of the findings

The findings are valid; some suggestions are provided below.

Additional comments

Kandelman et al. present data related to brainstem reflexes and clinical outcome from a binational, multicenter, prospective observational study on 137 brain-injured patients from the intensive care setting. Their main finding is that abolishment of the cough reflex within 24h after intubation predicted mortality, which remained significant after statistical adjustment for various confounding factors including sedation. This finding from brain-injured patients is in line with the same authors’ earlier data from non-brain injured patients.

The authors argue that, taken together, these results point towards brainstem dysfunction as an important and hitherto overlooked cause of mortality and morbidity in the intensive care unit. This is an intriguing and clinically highly relevant suggestion, which is likely to be picked-up by other researchers in this field.

The methods and statistics appear sound, the manuscript is well-structured (although the English could perhaps be improved somewhat), STROBE guidelines were followed, and the inferences made are sufficiently discussed, including most (but not all) limitations.

The overall conclusion is therefore that the present manuscript adds an important novel aspect to the intensive care literature.

However, I still have some points of criticism.

MAJOR CONCERNS

The authors discuss most limitations appropriately; the most important being lack of MRI assessing presence or absence of brainstem pathology and lack of EEG, respectively, sedative serum levels to assess the degree of sedation as the most significant confounder related to the abolishment of brainstem reflexes, including the cough reflex. However, although the possibility of brainstem dysfunction is certainly exciting, the data presented are inadequate to make inferences about causality – does brainstem dysfunction lead to multiorgan failure or vice versa? Obviously, these effects will almost certainly be bidirectional, but to which extent one or the other direction predominates cannot be answered by the data presented; this should be more clearly acknowledged. Also, the authors speculate that neuroinflammation may predispose to brainstem dysfunction, but is inflammation not rather unlikely to be a prominent factor after just 24 hours?

MINOR ISSUES

Although not strictly necessary, I think the manuscript would benefit from a detailed figure illustrating the concept of brainstem dysfunction, including the relationship between brainstem anatomy and pathophysiological mechanisms (and the bidirectional effects with systemic organ failure as outlined earlier).

Line 82-83: You may consider adding “… by reducing the rate of cerebral oxygen consumption and by decreasing intracranial pressure”.

Line 88: Should be “severely”

Line 89/90: Should be “…examination and electrophysiological testing”

Line 96: Should be “receiving deep sedation” or better “deeply sedated patients”

Lines 96-99: This statement must be backed up by a reference.

Line 103-104: Was the main objective not to extend the findings from non-brain injured critically ill
patients (reference 3) to brain-injured patients, rather than a mere confirmation?

Line 118: ”December 2011 and February 2015” – why the delay in reporting?

Line 131: Should be “Decisions…. mere made…” (decisions are not managed)

Line 152-153: Your previous results were based on non-brain injured patients, whereas your present study involved brain-injured patients, including those with lateralized structural brainstem lesions – in these patients, absence of unilateral (i.e. ipsilateral) brainstem reflexes might carry a worse prognosis. Please comment.

Line 154: Why not in TBI patients? Provided cervical spine injury has been excluded, VOR assessment is safe.

Line 155: Should be “Unresponsiveness”

Line 226: “neurological reasons” – please be more specific, and also state what the remaining reasons for intubation were.

Lines 246-248: How come that infratentorial lesions were associated with better survival? This seems counterintuitive, please comment.

Lines 325/326: Should be “limitations”

Line 345: This section should be marked as “Conclusions”.

Line 349: Should be “over-sedation”

---

## Round 0.2 · accepted · Accept

Thank you for your revised manuscript which has been accepted.

·

Basic reporting

-

Experimental design

-

Validity of the findings

-

Additional comments

I think the authors have adequately addressed all concerns. This is a very nice neurocritical care manuscript with an intriguing clinical finding. Congratulations.